An experimental search strategy retrieves more precise results than PubMed and Google for questions about medical interventions

Badgett Robert G. 1 rbadgett@kumc.edu
Dylla Daniel P. 2
Megison Susan D. 3
Glynn Harmon E 4
1 Department of Internal Medicine, Kansas University School of Medicine - Wichita , Wichita, KS , USA
2 Katy Campus Library, Houston Community College , Houston, TX , USA
3 No institutional affiliation , Birmingham, AL , USA
4 School of Information, University of Texas at Austin , Austin, TX , USA
Bentzen Søren
Electronic publication date: 2015 Apr 23
Publication date: 2015
Volume: 3
Electronic Location ID: e913
Received 2014 Nov 21; Accepted 2015 Apr 5
Copyright: © 2015 Badgett et al.
Copyright year: 2015
Copyright holder: Badgett et al.
License: This is an open access article distributed under the terms of the Creative Commons Attribution License, which permits unrestricted use, distribution, reproduction and adaptation in any medium and for any purpose provided that it is properly attributed. For attribution, the original author(s), title, publication source (PeerJ) and either DOI or URL of the article must be cited.
License URL: https://creativecommons.org/licenses/by/4.0/

Keywords: Information retrieval, Evidence-based medicine, Google, PubMed

Funding: The authors declare there was no funding for this work.

==============================
Objective. We compared the precision of a search strategy designed specifically to retrieve randomized controlled trials (RCTs) and systematic reviews of RCTs with search strategies designed for broader purposes.

Methods. We designed an experimental search strategy that automatically revised searches up to five times by using increasingly restrictive queries as long at least 50 citations were retrieved. We compared the ability of the experimental and alternative strategies to retrieve studies relevant to 312 test questions. The primary outcome, search precision, was defined for each strategy as the proportion of relevant, high quality citations among the first 50 citations retrieved.

Results. The experimental strategy had the highest median precision (5.5%; interquartile range [IQR]: 0%–12%) followed by the narrow strategy of the PubMed Clinical Queries (4.0%; IQR: 0%–10%). The experimental strategy found the most high quality citations (median 2; IQR: 0–6) and was the strategy most likely to find at least one high quality citation (73% of searches; 95% confidence interval 68%–78%). All comparisons were statistically significant.

Conclusions. The experimental strategy performed the best in all outcomes although all strategies had low precision.

Introduction

Health care providers are encouraged to answer clinical questions by first consulting evidence-based summaries (DiCenso, Bayley & Haynes, 2009). Summaries are defined as evidence-based practice guidelines and evidence-based textbooks. (DiCenso, Bayley & Haynes, 2009) Accordingly, physicians commonly use online resources such as UpToDate (UpToDate, 2014; Edson et al., 2010; Duran-Nelson et al., 2013).

Unfortunately, summaries may not always suffice. The evidence-based summaries UpToDate, Dynamed (DynaMed, 2014), FirstConsult (FirstCONSULT, 2014), and ACP Smart Medicine (American College of Physicians, 2014) have less than 5% overlap in the studies cited, which implies no resource is comprehensive (Ketchum, Saleh & Jeong, 2011). Similarly, studies report that UpToDate and the National Guidelines Clearinghouse addressed less than 80% of questions by primary care physicians (Fenton & Badgett, 2007) and hospital-based physicians (Lucas et al., 2004).

At times health care providers must search for original studies due to the deficiencies of secondary resources such as those discussed above; however, practicing physicians tend to have difficulty answering clinical questions by using electronic databases. This difficulty places physicians in the position of “knowing less than has been proved” (Mulrow, 1994). In a recent study, only 13% of searches by physicians led to changing provisional answers to correct while 11% of searches led to changing provisional answers to incorrect (McKibbon & Fridsma, 2006). Lucas found that 14% of inpatients were judged to have their care improved after physicians received unsolicited search results provided as part of a research study (Lucas et al., 2004).

The best search method for supplementing evidence-based summaries is controversial and difficult to identify due to the absence of a direct comparison of commonly used methods. The use of PubMed is encouraged by medical leaders; (PubMed, 2014i; AAMC-HHMI Scientific Foundation for Future Physicians Committee, 2009) however, physicians prefer the speed and simplicity of Google (Google, 2014a; Sim, Khong & Jiwa, 2008; Thiele et al., 2010). These methods fundamentally differ in the bibliographic data that are searched and in the sorting of search results. PubMed by default sorts results by date which may obscure a seminal article with more recent results from minor journals. On the other hand, Google, which sorts articles by a mix of estimated importance and relevance, ignores the dates of publication or revision of sources. Thus, Google may not accurately represent critical timing of search results that contain an article from a major journal that was later contradicted in a less impactful journal (Ioannidis, 2005). The implications of these differences are not fully known. Google launched Scholar in 2004 in order to improve access to academic publications. As compared to PubMed, Scholar indexes the full text of many journals rather than just the citation and abstract, but does not use MEDLINE’s metadata such as the National Library of Medicine’s Medical Subject Headings (MeSH) terms and publication types. Like Google web search, Google Scholar by default sorts citations by a mix of estimated importance and relevance. The retrieval algorithms and heuristics deployed by Google Scholar are propriety, not described on the Scholar website, and not clearly discernible (Google, 2014a).

In 1998, one of the authors (RGB) launched the experimental search engine SUMSearch, which includes PubMed searches and is specifically designed for use in clinical medicine to supplement evidence-based textbooks and practice guidelines (Badgett, 1999). The current version of SUMSearch is available at http://sumsearch.org. SUMSearch preserves the date sorting feature used by PubMed, but allows automated revisions of searches in order to make older sentinel articles visible. Automatic revisions of searches may address barriers health care providers experience while searching, such as designing search strategies and “uncertainty about narrowing the search...when faced with an overwhelming body of knowledge” (Ely et al., 2002).

Our objective is to quantify and compare the ability of a search designed specifically for clinical medicine with alternative strategies that are designed for broader purposes. We hypothesized that an experimental search strategy designed specifically for clinical topics would outperform other strategies for retrieving articles about medical interventions.

Materials & Methods

We compared five search strategies taken from four search engines for their ability to answer a collection of clinical questions. In previous comparisons, SUMSearch and PubMed have performed better than Scholar (Haase et al., 2007; Freeman et al., 2009; Anders & Evans, 2010); however, the current study is the first to compare SUMSearch and PubMed to each other and to Google. While Google and Google Scholar were not designed for clinical purposes, the frequency of their use by health care providers mandates assessment of their ability.

Source of clinical questions

We used questions about from the Clinical Questions Collection of the National Library of Medicine (National Library of Medicine, 2014; Ely et al., 1999; Ely et al., 2005). The complete collection consists of 4654 questions collected from physicians in Iowa. For each question, personnel at the National Library of Medicine assigned keywords that were almost always taken from the MeSH database.

From the collection we included questions about treatment of non-pregnant adults. This exclusion allowed us to better monitor development of the project as our clinical expertise is internal medicine. We excluded questions that also had a keyword assigned for diagnosis in order to ensure that the questions focused on treatment and so were best answered with randomized controlled trials and meta-analyses of trials. We excluded questions whose keywords duplicated the keywords of other questions. We included 312 questions after the above exclusions (Fig. 1).

Figure 1 Selection of questions.

Query expansion

Each question in the Questions Collection contains a median of 2 keywords, usually based on Medical Subject Heading (MeSH) terms. We linked these keywords with “AND.” In addition, we replaced the word “neoplasms” with “cancer” and inverted all keywords that contained commas. For example, “anemia, sickle cell” was inverted to “sickle cell anemia.” This inversion allows the search term to also perform well as a text word. The resulting search terms were submitted to the search engines without designation of a search field so that at PubMed’s Clinical Queries the terms were searched as both MeSH terms and text words. All searches were performed between June and December of 2009.

Search strategies

The experimental search strategy was based on the PubMed component of a prior version of SUMSearch federated search engine and could perform up to five iterations for each question. Details and examples of the iterations used by the experimental strategy are included in Table 2. This strategy sought randomized controlled trials and systematic reviews of trials. Each iteration was progressively more restrictive. The composition and sequencing of the iterations was based on experience with SUMSearch. The strategy returned the results of the last iteration that retrieved 50 or more citations. The rationale for restricting the numbers of citations is to reproduce the behavior observed in searchers to typically scan a limited number of citations (Blair, 1980; Islamaj Dogan et al., 2009). This limit has been called the futility point and occurs when searchers regard reviewing additional citations as being beyond their respective time and manageability constraints. The experimental search strategy imitated PubMed searching by querying Entrez’s eSearch utility (Sayers, 2014). This utility has no user interface and is designed by the National Library of Medicine for external search engines and other automated tools to efficiently query PubMed.

Table 1 Example clinical question and resulting search strategy.

Original question by primary care physician	“If someone had x-rays for acne treatment, how should they be followed-up regarding thyroid cancer risk?”	
Keywords originally assigned by the Clinical Questions Collection	Thyroid neoplasms Radiation Injuries	
Search submitted to PubMed’s Clinical Queries (Therapy category) and to Experimental search	Thyroid cancer AND Radiation Injuries	
Search submitted to Google and Scholar*	Thyroid cancer Radiation Injuries PMID ∼random ∼trial	
Notes.

* For users to reproduce the strategies with the current version of Google, settings are configured for “Google Instant Predictions” to be off and Results per page to be 50. The tilde signs are no longer required by Google as Google currently searches for synonyms by default. Since execution of our study, Google has revised Scholar to allow a maximum of 20 results per page.

Table 2 MEDLINE iterations of the experimental search strategy.

Iteration	Options to increase specificity of search	
	Quality filters	Publication types	Additional query expansion	
1	No filter	None	None	
2	Haynes 2005 sensitive filter* or systematic review subset	Excluded publication type of review, letter, editorial	Required abstract	
3	Switched to Haynes 2005 specific filter or systematic review a	No change	No change	
4	Added restriction to 106 journals in McMaster list as of 02/2008 b	No change	No change	
5	No change	No change	Added restriction of search terms to MeSHc major field	
Notes.

a Filters are detailed in the original study by Haynes et al. (2005) and at http://hiru.mcmaster.ca/hiru/HIRU_Hedges_home.aspx.

b Journals are listed at http://hiru.mcmaster.ca/hiru/hedges/MedlineJournalsRead.pdf.

c Medical Subject Headings terms assigned by the National Library of Medicine.

We included two strategies from PubMed’s Clinical Queries that are publicly available (http://www.ncbi.nlm.nih.gov/pubmed/clinical). We used the current Narrow and Broad strategies for therapies. These strategies were initially developed by Haynes in 1994 and revised by Haynes in 2005 (Haynes et al., 1994; Haynes et al., 2005).

We studied two strategies by Google. We used the main Google Web search engine and labeled this strategy as “Google.” We used the Google Scholar search engine and labeled this strategy as “Scholar.” For both of these strategies, we assessed methods to improve upon simply constructing search queries by using clinical terms. Using test cases, we informally assessed the benefit of adding the following candidate search terms to the search query: “PMID,” “DOI,” ∼random, ∼trial, site:.org, site:.edu, and site:.gov. The terms PMID and DOI are abbreviations for “PubMed identifier” and “digital object identifier” and are common numeric identifiers in the Internet addresses and on the Internet pages for articles in health care journals. These identifiers are indexed by Google like any other content on an Internet page or in its Internet address. In addition, formal citations to health care articles, such as in wikis, frequently include these numbers and the abbreviations that indicate the type of number. The final strategy chosen for both Google and Scholar appended the strings “PMID,” “∼random,” and “∼trial” to the search terms. The “∼” character was required at the time of our study for Google to seek synonyms for an adjacent search term (Schwartz, 2014). We appended “num=50” to the urls submitted for both strategies in order to retrieve 50 hits per search. Searches were performed on a dedicated server that had Google cookies removed in order to prevent Google from any customization of search results such as prioritizing results based on geographic location.

Outcome ascertainment

All search results were parsed for PMIDs and DOIs. For search results from Google, we also parsed the text in the Internet addresses of hyperlinks. All identifiers found were then submitted to Entrez’s efetch utility in order to retrieve full citations including PMIDs, MeSH terms and lists of all articles that commented on the retrieved articles.

Reference standard

The reference standard required articles to be relevant and high quality. An article was considered relevant to the clinical question if the article contained all of the keywords assigned by the Clinical Questions Collection to the clinical question. The keywords could be either MeSH terms or MeSH entry terms, and the keywords could be located in title, abstract, or MeSH terms of the article.

An article was considered high quality if it had high quality methodology or was considered important by an expert in the domain of the article. Articles having high methodological standards were considered those that were reviewed by an evidence-based synoptic journal as previously done by Aphinyanaphongs (Aphinyanaphongs et al., 2005). These journals were ACP Journal Club, InfoPoems, Evidence Based Dentistry, Evidence Based Medicine, Evidence Based Nursing, and Evidence Based Mental Health. Articles considered important by a domain expert were those that were published with an accompanying editorial.

To avoid incorporation bias that would artificially inflate our estimated of the accuracy of the searches, all strategies were designed without incorporating search terms that contribute to the definition of the reference standard. For example, one component of our reference standard is abstraction of the article by the publication ACP Journal Club. Some websites, such as PubMed, indicate which citations have been reviewed by ACP Journal Club. Thus, we could have added “ACP Journal Club” to our search strategy to improve its precision. However, we did not add this term, as it would create incorporation bias and limit the ability to generalize the results of our study to topics not covered by ACP Journal Club. An example question from the Clinical Questions Collection and the resulting search strategies is in Table 1.

Statistical analysis

The primary outcome was the median average precision of the searches for retrieving studies meeting criteria for the reference standard. We limited the number of search results examined to 50 to control for the varying number of results retrieved by each search engine. For example, searches for medical interventions may retrieve hundreds of thousands of results using the Google strategy while retrieving a much smaller number with the other search strategies. We specified 50 search results because searchers, outside of those performing meta-analysis, are unlikely to review a large number of citations (Blair, 1980; Islamaj Dogan et al., 2009) In addition, this limit allows comparison of searches that may retrieve substantially different number of citations (Herskovic, Iyengar & Bernstam, 2007). For example, Google may retrieve more citations of high quality than the other strategies due to retrieving many-fold more total citations. However, the Google search is not clearly better because the user had to sift through more citations to find the high quality citations.

The precision was calculated as the proportion of the first 50 search results identified by each strategy that were deemed to be relevant, high quality studies according the criteria in the preceding section, “Reference standard.” If no qualifying studies were retrieved, the precision was set to 0.

The number need to read (NNR) for each strategy is the number of citations that would have to be assessed to yield one qualifying article. The NNR was calculated as the inverse of the precision (Toth, Gray & Brice, 2005).

Calculations were made with R statistical software package, version 2.11.1 (R Development Core Team, 2012). Pairwise comparisons between individual medians were assessed using a post hoc analysis for Friedman’s Test. Rates of dichotomous outcomes were compared with the chi-square test. Chi-square is a conservative choice as it does not consider pairing of data in calculation.

Results

The most common clinical concepts in the 312 questions about treating non-pregnant adults were hyperlipidemia (15 questions), hypertension (10 questions), and urinary tract infections (10 questions).

The principal outcome, search precision, and all other outcomes were not normally distributed (Lilliefors normality test p < 0.001), so the median precision became the principal outcome. Using Google as an example to illustrate the results, when the first 50 hits in a Google search were examined, a mean of 23 PubMed citations were retrieved by parsing PMIDs or DOIs from the Google results (not shown in table). Of these 23 PubMed citations, an average of 3.3 were deemed high quality because the citation was abstracted by an evidence-based synoptic journal or published with an accompanying editorial. Of the 3.3 citations, an average of 1.3 was relevant to the original search terms. While this suggests the mean precision for Google was 1.3 divided by 23, or 5.6%, the actual mean precision was lower at 4.6%. The discrepancy is because the average of a series of fractions is not equivalent to the average of the numerators divided by the average of the denominators. Lastly, 54% searches performed by Google retrieved no high quality, relevant citations thus the median precision for Google was 0% (Table 3). The corresponding values for the numbers needed to read are: Experimental 18, PubMed narrow Clinical Query 25, and PubMed’s broad Clinical Query 50. The numbers needed to read cannot be calculated for the Google strategies due to their median precision of 0%.

Table 3 Comparison of search strategies for retrieving high qualitya, relevant PubMed citations.

Experimental	PubMed clinical queries for therapies	Google	Google scholar	
	Narrow	Broad			
Precision of searches, median (interquartile range) b , c	
5.5%d	4.0%d	2.0%	0%d	0%d	
(0% to 12%)	(0% to 10%)	(0% to 8%)	(0% to 7%)	(0% to 0%)	
Number of citations retrieved, median (interquartile range) b , c	
2d	2	1	0d	0d	
(0 to 6)	(0 to 4)	(0 to 3)	(0 to 2)	(0 to 0)	
Proportion of searches that retrieved at least one citation (95% confidence intervals)b	
73%d	63%	65%	46%d	20%d	
(68% to 78%)	(58% to 68%)	(59% to 70%)	(41% to 52%)	(15% to 24%)	
Notes.

a High quality citations were those reviewed by an evidence-based synoptic journal or accompanied by an editorial.

b P < 0.001 for differences among groups.

c Note that rank sums can differ significantly although medians are the same.

d P < 0.05 for difference compared to other groups.

Search results were limited to a maximum of 50 per search.

The median precision was significantly different among the strategies by Friedman’s rank sum test (Table 3). The experimental strategy and the narrow strategy of the PubMed Clinical Queries had the highest median precision (5.5% and 4.0%, respectively). The experimental strategy had the highest ranked and mean values of precision (Table 3; p < 0.001 for both analyses). The experimental strategy was the most likely method to find at least one high quality citation (73% of questions) with p < 0.001. The median number of high quality articles retrieved per search was two for both the experimental strategy and the PubMed narrow, while the means were 5.0 and 2.6, respectively (p < 0.001).

In an unplanned analysis, we examined the precision of experimental search strategies based on the number of iterations the experimental strategy required (Fig. 2). Searches that required one or two iterations had low precision, whereas searches requiring more iterations had higher precision.

Figure 2 Precision by number of interations used by the experimental search engine.

For all outcomes, Google and Google Scholar performed worse than the other strategies. This was in part because Google itself sometimes found high quality citations that were not relevant. For example, in a search for bronchiectasis and drug therapy, Google retrieved the Wikipedia pages on acetylcysteine and pulmonary embolism. The acetylcysteine page was retrieved because Wikipedia listed bronchiectasis as treatable with acetylcysteine while the pulmonary embolism page was retrieved only because the page listed bronchiectasis in the page’s navigational menu of pulmonary diseases. Unfortunately, the high quality citations that were on these pages were not relevant to bronchiectasis.

Discussion

The experimental search was significantly better than the other strategies in all outcomes. Google and Google Scholar strategies did not perform as well. We believe this is the first comparative study to identify a search strategy that may be comparable to or better than the 2004 version of the PubMed Clinical Queries for common clinical questions. The experimental search is available at http://sumsearch.org/ by changing the default settings so that “Focus” is Intervention, “Target # of original studies” is 50, and “Require abstracts” is not selected.

Our results support Battelle’s hypothesis that domain-specific search strategies should perform better than general strategies (Battelle, 2005). Google and Google Scholar’s poor performance was consistent with prior comparisons with PubMed or SUMSearch (Haase et al., 2007; Freeman et al., 2009; Anders & Evans, 2010). Our study should be compared to three studies that suggest benefit from using Google Scholar. Gehanno notes perfect coverage by Scholar of trials in a set of Cochrane reviews (Gehanno, Rollin & Darmoni, 2013). However, coverage simply relates to the presence of trials in the Scholar database, which is different from our study of how well those trials can be retrieved by search strategies. Two smaller studies, by Nourbakhsh and Shariff suggests that Scholar retrieves more citations that are relevant than PubMed retrieves (Nourbakhsh et al., 2012; Shariff et al., 2013). Several reasons may underlie the conflicting results. The reference standard used by Nourbakhsh only considered relevance and not study design or quality of citations. Our precision is likely underestimated due to the certain existence of qualifying articles that were missed due to not being abstracted by an evidence based synoptic journal. Also, the PubMed searches used by Nourbakhsh relied exclusively on MeSH terms. (Nourbakhsh et al., 2012) For example, Nourbakhsh used “hypertension, pulmonary [MeSH]“ whereas we would have changed this term to “pulmonary hypertension[all fields].” The ‘all fields’ tag submits the term as both a MeSH term and a text word. In addition, the Nourbakhsh study was limited to four questions the researchers were familiar with and the differences did not reach statistical significance. Shariff did not provide details on how the nephrologists used PubMed other than stating that the searches were not revised based on the number of results retrieved (Shariff et al., 2013). The findings of similar precision of results yet fewer relevant citations among the first 40 citations retrieved by PubMed compared to Scholar indicates that in many cases the PubMed searches retrieved fewer than 40 citations. The conflict between our results and those of Shariff may be due to our use of iterative searching or to the nature of primary versus specialty care questions. Iterative searching may be more important in broad topics that retrieve more citations.

The domain-specific search strategies that we studied, PubMed and SUMSearch, may perform better for two reasons that have not changed since our study was completed. First, these strategies, unlike Google and Scholar, take advantage of the hierarchal Medical Subject Headings (MeSH) terms that the National Library of Medicine assigns to citations. Second, our results raise the question of whether a Boolean search model should be preferred for the task we studied. Most contemporary research of searching MEDLINE examines search models other than Boolean. Boolean models connect search terms with logical connectors such as ‘and’ and ‘or’ are considered weaker than other search models (Baeza-Yates & Ribeiro-Neto, 2011). A paradoxical advantage of Boolean models is that because they do not rank documents by any grading scale, search results can be sorted by date of publication. Sorting by date can be critical in medicine because of the surprising frequency that research results are contradicted by subsequent authors (Ioannidis, 2005).

In addition to providing a comparison of the performances of commonly used search strategies, our results reinforce the difficulty of retrieving clinical studies from MEDLINE. The experimental strategy was most precise but barely achieved a precision of 5%. Our study reported substantially lower precision than a previous comprehensive review by McKibbon (McKibbon, Wilczynski & Haynes, 2009). Common to our study and the review was analysis of the PubMed Clinical Queries narrow filter. McKibbon reported a precision of 55% whereas we found the same filter to have a precision of 2%. We believe our study reflects the precision that health care providers will encounter and is lower than the report of McKibbon for two reasons. First, we measured the precision in answering actual clinical questions. Second, we measured the precision among all journals of PubMed rather than limiting to the 161 journals that publish the highest rate of high-quality studies. Since we executed our study, Shariff reported that nephrologists were able to search MEDLINE with a mean precision higher than our report of median precision (Shariff et al., 2013). We reported median rather than mean values for precision due to concern that means will overstate performance. To directly compare studies, the mean precision of 10.2% we report for our experimental strategy is higher than found by Shariff.

Possible limitations

First, we standardized the design of all search strategies to eliminate variability in the search skill of actual users. Both the precision and number of relevant citations retrieved by human searchers may be less than we report. It is possible that in our study Google’s performance was diminished because Google may have found citations that were not counted because they were not accompanied by PMIDs or DOIs. However, in addition to parsing the results displayed by Google, we also parsed the links provided by Google. Any functional link to an article at PubMed will have a PMID embedded and be found by our methodology. Similarly, high quality studies may have been missed by all strategies due to our removing “diagnosis” as a key word. This may have selectively harmed the experimental and PubMed strategies as these incorporated MeSH terms. However, it is unknown whether this affected precision as the total number of studies retrieved is also lower.

We recognize that our definition of the reference standard might be debatable for three reasons. First, we limited our study to retrieving randomized controlled trials and systematic reviews of randomized controlled trials because treatment questions are important and the standards for the conduct and assessment of these studies are better developed than for other resources. While this information need may be infrequent for many health care providers, we believe the ability to locate randomized controlled trials is very important for peer leaders who may be writing or teaching clinical topics. Second, our definition of high quality articles is imperfect. We believe, however, our definition has the advantage of being determined by experts who determined that an editorial or synopsis was justified and who were not involved in the evaluation of the search strategies. In addition, we believe the results that our definition yields are likely to move in parallel with other definitions of high quality. Third, the use of precision (the proportion of relevant documents retrieved in the search) as a metric instead of sensitivity (the ability of the system to detect all relevant documents) is debatable. However, our goal is to create searches with precision for clinicians rather than comprehensive searches for meta-analysts. For example, high sensitivity may be more useful for meta-analyses that require comprehensive results. High precision may be more useful for time-sensitive tasks that require relevant documents quickly. Regardless, we do provide the numbers of high quality citations retrieved which should correlate with the sensitivity of a strategy for a given question.

Our results should not be generalized beyond searching for studies of interventions. The randomized controlled trial index term used by the National Library of Medicine‘s Medical Subject Headings (MeSH) is unusually accurate whereas MeSH terms for other study designs may be less accurate (Haynes et al., 2005). None of the strategies we tested may be appropriate for the conduct of meta-analysis when very high recall or sensitivity of searches is required. Lastly, our questions all had carefully assigned MeSH terms. Searchers not facile with MeSH terms may have lesser results.

Future research

Future research could address the strategies that were studied and compare them to search strategies based on alternative search models. Aside from the search strategies developed by Haynes for PubMed’s Clinical Queries, the strategies were not formally developed. For example, we appended Google and Scholar strategies with “PMID ∼random ∼trial” based on several use cases, but perhaps other restrictions would have performed better. However, Google’s performance was so low that substantial improvement from revising search terms seems unlikely. Google frequently revises its search algorithms (R Development Core Team, 2011). Until Google makes a major change, such as recognizing MeSH terms and the hierarchical relationship among them, the impact of lesser revisions on searching for medical research is not known. Continual research of Google is warranted. Regarding the experimental strategy, perhaps other iterations, sequences of iteration, and number of iterations would improve the search results. In addition, Wilczynski recently described how to improve the precision of the Haynes strategies by adding “not” terms to searches of MEDLINE (Wilczynski, McKibbon & Haynes, 2011) Future research could compare our strategy to strategies based on machine learning or citation analysis. Lastly, we hope that search engines in the future will provide more than a list of citations and will add indicators of credibility to citations and display the conclusions in a way to allow users to quickly assess the concordance among conclusions. The former is currently done by SUMSearch by indicating which citations are accompanied by editorials and reviews by synoptic publications. The latter is being developed by AskHermes (Yu, 2014).

Conclusion

Our results suggest that when health care providers need to supplement evidence-based summaries by searching for high quality randomized controlled trials and systematic reviews of randomized controlled trials, an experimental strategy designed specifically for clinical care may be more appropriate than the more general strategies deployed by Google and PubMed Clinical Queries.

Additional Information and Declarations

Competing Interests

Author Contributions

Dr Badgett created SUMSearch, which is the basis for the experimental search strategies. Dr Badgett receives no compensation for this project or for SUMSearch.

Robert G. Badgett conceived and designed the experiments, performed the experiments, analyzed the data, wrote the paper, prepared figures and/or tables, reviewed drafts of the paper.

Daniel P. Dylla and Susan D. Megison performed the experiments, wrote the paper, reviewed drafts of the paper.

E Glynn Harmon reviewed drafts of the paper and provided oversight.

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
