# Peer review of "An experimental search strategy retrieves more precise results than PubMed and Google for questions about medical interventions"

_PeerJ, doi:10.7717/peerj.913_

## Round 0.1 · original submission · Major Revisions

· Academic Editor

Major Revisions

I concur with the reviewers that your paper is concerned with an important issue but that the limitations to the study methodology and the potential impact on the conclusions should be discussed in more detail.

·

Basic reporting

This paper is well written, properly structured, and has excellent clear presentation throughout the introduction. My only quibble is that I do think line 265 ”Boolean models connect search terms with logical connectors such as “and” and “or” are considered weaker than other search models.” does not fit. I am familiar with some of Baeza-Yates older work, not the paper cited, so I am not questioning that there is something to this statement, and I agree with line 264-265 however, making such a statement without further elaboration in a paper where you feel it necessary to explain what Boolean searching is, seems out of place.

Experimental design

Inclusion criteria: I would like to see a more detailed explanation of how the subset of 367 of 4654 questions was derived. See for example the explanation in Ketchum AM, Saleh AA, Jeong K. Type of evidence behind point-of-care clinical information products: a bibliometric analysis. J Med Internet Res. 2011 Feb 18;13(1):e21. doi: 10.2196/jmir.1539. PubMed PMID: 21335319; PubMed Central PMCID: PMC3221343.

Why non-pregnant adults? Because there are fewer RCTs on pregnant adults and children? The inclusion criteria may be sound, but they require justification.

Some of the reference standard sources work from the list of journals used in iteration 4 of the experimental search strategy. Would this not boost the precision of the experimental strategy?

Validity of the findings

I question if the Google Scholar results are valid. First, they don’t fit with my experience from searching for clinical questions, they don’t fit with Nourbakhsh or Shariff (both cited) and finally, my own limited testing with using an article title vs using an article title and appending “PMID” show very clean results, and a PubMed record was one of 7 similar grouped together in the first record returned. However, appending “PMID” to the same search string resulted in a relative mess. This could be due to a change since the experiments were run in 2009, which would invalidate results. It may be that the authors did fully explore the impact of adding “PMID”. It may be that the authors have tested the method and it works well, but this would need more detailed reporting in the methods section. The Google results may be off as well. Either way, searching Google to find PubMed results seems less than optimal approach, and that seems to have been the goal. On this theme, I wonder why the domain ncbi.nlm.nih.gov wasn’t tested as part of the Google searches, rather than PMID. But if the functional links were being searched, linked references may still have be picked up (as in the Wikipeda examples, lines 227 and 228)
On that note, I don’t see any report of Google Scholar in lines 210-214.
This paper may be stronger without the Google/Google Scholar portion, or the methods may simply require greater explanation.


I’m not convinced all high quality articles are reviewed by an evidence based synoptic journal or have editorials written about them. These would be better indicators of clinical importance, that is, those articles that are of high quality and are likely to change practice. ACP Journal club, for example provides lists of article that meet their quality standards but were not also considered.
The precision found is barely better than that found for systematic review searches for interventions (median 0.027) and worse than the “other” category (median 0.065). (Sampson, M., Tetzlaff, J., & Urquhart, C. (2011). Precision of healthcare systematic review searches in a cross-sectional sample. Research Synthesis Methods, 2(2), 119–125. doi:10.1002/jrsm.42) Give that the systematic review searches actively trade precision for recall, and the precision is based on searches of multiple databases, it is hard to believe that the reference standard is complete.

Comments for the author

I'm a little worried about the reference standard, but if you can address those concerns, I think you've got a great paper here.

Minor points:
http://clinques.nlm.nih.gov/ may no longer be available – I was not able to access it on December 3 or December 10, 2014.

·

Basic reporting

The article is well written and the text is relatively easy to follow.

The topic is important, but I think the introduction and discussion could be widened in scope as follows:

1) I would recommend mentioning that the diffusion/adoption of new medical interventions is relatively slow as a motivation for improving the access to high quality knowledge for the individual clinician through higher precision searches (with some reference; I found "Nation-scale adoption of new medicines by doctors: an application of the Bass diffusion model" as a first example of the issue, but I think there must be more relevant studies out there)

2) I think that complicated access models to subscription model journals is an issue that deserves mentioning as well, as I suspect it is part of the reason for the relatively low popularity of Pubmed referenced in line 62/63. I believe this may be worthwhile mentioning.
I asked my better half (MD at residency level) where she would go for information first and got the answer "google (or upToDate)". When I asked why not pubmed she answered "because you need a subscription to view the results". Paradoxically, the hospitals WILL have the appropriate subscriptions, but only through an awkward login and redirections. Now, I am not suggesting that you should add a long section, but a single sentence pointing to the problem may be relevant.

Specific suggestions:

Line 202-210: The example of non-normal distribution of the Google data: This is a relatively long discussion of an issue that is not really rocket science. I suggest cutting it down.

Line 310-319: I think you 'sit bewteen two chairs': Your study aims at providing high precision searches, not comprehensive (high sensitivity) searches for meta-analysis. Couldn't you state it that simply? Line 310-319 appears (to me) as an attempt to 'salvage' a measure correlated with sensitivity and then backpaddle.

Line 302-304: I strongly believe that randomized evidence IS important, even to the practicing clinician!

I would suggest removing the subheadings in the 'discussion' section.

A couple of typos:

Line 108: MESH abbreviation is already introduced, so no need to spell out
Line 251: 2x "in addition"
Line 252 "did reach not"

Experimental design

The research question is well defined, relevant and meaningful.

Some questions regarding the design and methods:

How did you select the 367 questions (line 101)? Please specify

I am concerned that the approach of looking at 50 'hits' from Google and then reducing to 'PMID's (23 in your example line 208) gives you an unfair median precision of Google: Shouldn't you have restricted to 50 PMID's instead to compete fairly with Pubmed? (I guess the individual precision is OK if you find at least one relevant and high quality citations, but you will have too many 0-precision searches?). I realize you mention it as a possible limitation in the discussion, but isn't it more than 'possible'?

Regarding the "Reference standard" (line 153-164):

I realize your method requires a machine-interpretable measure of quality and relevance. However, your approach is intended to improve the precision of searches for clinicians and you make no attempt to correlate your measure of quality and relevance to the needs of a human search machine user (clinician). I would have greatly appreciated if a human observer had tried at least some searches, so you could document if a 'relevant and high quality' reference according to your definition agrees with the need of a 'real' clinician.

Second, I am concerned whether the definition requiring "reviewed by an evidence-based synoptic journal" would lead to a timing bias, excluding high-quality articles of recent publication date?


As a general comment, I believe you fail to address the RANKING of the 50 results. The number 50 is somewhat arbitrary - some clinicians will probably look no further than the first 20. Therefore I think the ranking of the search results is important. Maybe a more relevant definition of 'number needed to read' would be median number of hits to scan before the FIRST relevant and high quality citation?

Validity of the findings

I am satisfied with the validity of the findings, albeit the methodological issues pointed out above should be addressed (or defended).

---

## Round 0.2 · Minor Revisions

· Academic Editor

Minor Revisions

Please address the remaining issues from the two reviewers. I concur with the reviewers that these are important for a potential reader. I am also wondering about the change in the number of questions in the revised manuscript.

·

Basic reporting

p.5 Source of clinical questions
I don't find it any easier to understand how 4645 questions dropped to 367 and then 312 in this revision. I would like a clear statement of inclusion criteria and a CONSORT or PRISMA-style flow diagram accounting for the exclusions.

Experimental design

No comments or concerns.

Validity of the findings

p. 6 "The experimental search strategy was based on the PubMed component of a prior version of SUMSearch." Please cover, possibly in the discussion as a limitation, how the current version of SUMSearch differs and what impact this may have on the findings. State if the changes are such that there would be no impact.

Comments for the author

These are all minor points:
p. 6 bottom and p.16 final sentence – do you mean clinical queries here, rather than clinical studies?

p.11 The first sentence of the discussion is incomplete … better in all outcomes than what?

p. 12 “Most contemporary research of searching MEDLINE examines search models other than Boolean.” I respect that you took this out to simplify, but it sums up the current state of the art very nicely and I was going to cite this! Consider re-introducing this sentence.

p. 14 "However, our goal is to create searches with precision for busy clinicians rather than comprehensive searches for meta-analysts.” The term “busy clinicians” is a cliché – please use some other phrase here.

·

Basic reporting

Some of the in-line references appear to be slightly odd format with '. ,': Duran-Nelson et al. , 2013 ; Please check.

Experimental design

Regarding the "Number needed to read": I think I did not express myself clearly in the last iteration. You choose a definition of 'number needed to read' that is 1/precision, which has been used before in the literature:

"The number need to read (NNR) for each strategy is the number of citations that would have to be assessed to yield one qualifying article. The NNR was calculated as the inverse of the precision. (Toth, Gray, & Brice , 2005)"

My question was: Is this a good definition of Number needed to read? An ALTERNATIVE definition would be the number of results to browse before the first relevant and high quality citation.
Lets imagine two searches of 50 hits with 2 relevant and high quality results each, leading to NNR=25 for both with your method (right?). If it is number 5 and 7 in search strategy one that are high quality and no 35 and 42 in search 2, I would expect a 'number needed to read' definition to distinguish the two searches.
My homemade suggestion would be that NNR=5 and 35 instead. Does that make sense? I am not insisting, but I think it is a better reflection of real life.

Validity of the findings

I believe the rebuttals from the authors are sufficient.

---

## Round 0.3 · accepted · Accept

· Academic Editor

Accept

Thank you for addressing the comments by the reviewers.